# No evidence for prolactin's involvement in the post-ejaculatory refractory period

Susana Valente [1,2], Tiago Marques [3,4,5] & Susana Q. Lima [1 ✉]

In many species, ejaculation is followed by a state of decreased sexual activity, the post-ejaculatory refractory period. Several lines of evidence have suggested prolactin, a pituitary hormone released around the time of ejaculation in humans and other animals, to be a decisive player in the establishment of the refractory period. However, data supporting this hypothesis is controversial. We took advantage of two different strains of house mouse, a wild derived and a classical laboratory strain that differ substantially in their sexual performance, to investigate prolactin's involvement in sexual activity and the refractory period. First, we show that there is prolactin release during sexual behavior in male mice. Second, using a pharmacological approach, we show that acute manipulations of prolactin levels, either mimicking the natural release during sexual behavior or inhibiting its occurrence, do not affect sexual activity or shorten the refractory period, respectively. Therefore, we show compelling evidence refuting the idea that prolactin released during copulation is involved in the establishment of the refractory period, a long-standing hypothesis in the field of behavioral endocrinology.

[1] Champalimaud Research, Champalimaud Centre for the Unknown, Av. Brasilia, s/n Lisboa, Portugal. [2] Graduate Program in Areas of Basic and Applied Biology (GABBA), University of Porto, 4200-465 Porto, Portugal. [3] Department of Brain and Cognitive Sciences, MIT, Cambridge, MA 02139, USA. [4] McGovern Institute for Brain Research, MIT, Cambridge, MA 02139, USA. [5] Center for Brains, Minds and Machines, MIT, Cambridge, MA 02139, USA. ✉email: susana.lima@neuro.fchampalimaud.org

Sexual behavior follows the classical sequence of motivated behaviors, terminating with an inhibitory phase after ejaculation: the post-ejaculatory refractory period (PERP)[1]. The PERP is highly conserved across species and includes a general decrease in sexual activity and also inhibition of erectile function in humans and other primates[2]. This period of time is variable across and within individuals and is affected by many factors, such as age[3,4] or the presentation of a new sexual partner[5,6]. The PERP is thought to allow replacement of sperm and seminal fluid, functioning as a negative feedback system where by inhibiting too-frequent ejaculations an adequate sperm count needed for fertilization is maintained[7,8].

Several lines of evidence have suggested the hormone prolactin (PRL) to be a key player in the establishment of the PERP[9,10]. PRL is a pleiotropic hormone, first characterized in the context of milk production in females, but for which we currently know several hundred physiological effects in both sexes[11,12]. The association of PRL to the establishment of the PERP in males is based on several observations. First, it was shown that PRL is released around the time of ejaculation in humans and rats[13–21]. Anecdotally, no PRL release has been observed in a subject with multiple orgasms[22]. Second, chronically abnormal high levels of circulating PRL are associated with decreased sexual drive, anorgasmia, and ejaculatory dysfunctions[23,24]. Finally, removal of PRL-producing pituitary tumors or treatment with drugs that inhibit PRL release reverse sexual dysfunctions[25,26]. Taking these observations into consideration, it has been hypothesized that the PRL surge around the time of ejaculation plays a role in the immediate subsequent decrease of sexual activity, the hallmark of the PERP. In fact, this idea is widespread in behavioral endocrinology textbooks[27] and the popular press (https://en.wikipedia.org/wiki/Refractory_period; https://www.humanitas.net/treatments/prolactin).

PRL is primarily produced and released into the bloodstream from the anterior pituitary[11,28] and consistent with its functional diversity, PRL receptors are found in most tissues and cell types of the body[29,30]. Therefore, PRL may depress sexual activity directly, via PRL receptors present in the male reproductive tract. In fact, PRL has been shown to impact the function of accessory sex glands and to contribute to penile detumescence[31]. PRL can also affect central processing, as it can reach the central nervous system either via circumventricular regions lacking a blood–brain barrier[32] or via receptor-mediated mechanisms[33], binding its receptors which have widespread distribution, including in the social brain network[34]. Hence, circulating PRL can impact the activity of neuronal circuits involved in the processing of socio-sexual relevant cues and thus sexual performance. Circulating PRL reaches the central nervous system on a timescale that supports the rapid behavioral alterations that are observed immediately after ejaculation (in <2 min)[35]. Through mechanisms that are not yet well established, PRL is known to elicit fast neuronal responses[36] besides its classical genomic effects[37]. In summary, circulating PRL can impact several systems involved in sexual behavior on a timescale compatible with the establishment of the PERP.

However, despite data supporting the involvement of the ejaculatory PRL surge in the establishment of the PERP, this hypothesis has received numerous critics[2,3,38–40]. While in humans it is well established that chronically high levels of PRL reduces libido[24], some authors suggest that those results were erroneously extended to the acute release of PRL around ejaculation[2,3,38–40]. Furthermore, there is controversy in relation to PRL dynamics during sexual behavior, since in most studies PRL levels were quantified during fixed intervals of time, and not upon the occurrence of particular events, such as ejaculation. In fact, some reports in rats suggest that PRL levels are elevated through the entire sexual interaction[41,42]. Finally, formal testing of the impact of acute PRL manipulations on sexual activity and

performance is still missing (but see ref. [43] for an acute manipulation in humans).

In the present study, we tested the role of PRL in sexual activity and in the establishment of the PERP in the mouse. The sequence of sexual behavior in the mouse is very similar to the one observed in humans[44], making it an ideal system to test this hypothesis. Also, we took advantage of two strains of inbred mice that are representative of two different mouse subspecies (C57BL/6J: laboratory mouse, predominantly *Mus musculus domesticus* and PWK/PhJ: inbred wild-derived, *Mus musculus musculus*[45]) and exhibit different sexual performance. Through routine work in our laboratory, we observed that while most BL6 males take several days to recover sexual activity after ejaculation, a large proportion of PWK males will re-initiate copulation with the same female within a relatively short period of time. This difference in PERP duration can be taken to our advantage, widening the dynamic range of this behavioral parameter and increasing the probability of detecting an effect with pharmacological manipulations.

First, we show that there is PRL release during sexual behavior in male mice. Second, using a pharmacological approach, we show that acute manipulations of prolactin levels, either mimicking the natural release during sexual behavior or inhibiting its occurrence, do not affect sexual activity or shorten the refractory period, respectively.

## Results

**Prolactin is released during sexual behavior in male mice**. We first asked if PRL is released during copulation in the two strains of male mice. To monitor PRL dynamics during sexual behavior we took advantage of a recently developed ultrasensitive ELISA assay that can detect circulating levels of PRL in very small volumes of whole blood (5–10 μl), allowing the assessment of longitudinal PRL levels in freely behaving mice[46]. Sexually trained laboratory mice (C57BL/6J, from here on BL6) and inbred wild-derived mice (PWK/PhJ, from here on PWK) were paired with a receptive female and allowed to mate (see "Methods" for details). During the sexual interaction males were momentarily removed from the cage to collect tail blood after which they returned to the behavioral cage, resuming the sexual interaction with the female. We collected blood samples upon the execution of pre-determined, easily identifiable, behavioral events that correspond to different internal states of the male: before sexual arousal (baseline, before the female was introduced in the cage), at the transition from appetitive to consummatory behavior (MA, mount attempt, immediately after the male attempted to mount the female), during consummatory behavior (mount, after a pre-determined number of mounts with intromissions, BL6 = 5 and PWK = 3), and after ejaculation (ejaculation, after the male exhibited the stereotypical shivering, falling to the side and separating from the female) (Fig. 1a, see "Methods" for details).

Baseline levels of circulating PRL in male mice were low for both strains (BL6 0.86 ± 0.46; PWK: 2.31 ± 1.37 ng/ml; please see ref. [46] for BL6), but are significantly increased during sexual interaction (Bl6: $F_{3,7} = 21.26$, $P < 0.0001$; PWK: $F_{3,8} = 17.18$, $P < 0.0001$; RM one-way ANOVA) (Fig. 1a). While in the case of BL6 males PRL levels only increased during the consummatory phase, PRL levels in PWK males are significantly increased already at the transition from appetitive to consummatory behavior (Baseline vs MA 16.30 ± 6.67 ng/ml, $P = 0.001$, Tukey's multiple comparisons test) (Fig. 1a). In both strains, PRL levels after ejaculation are similar to the levels reached during consummatory behavior (BL6 $P = 0.71$ vs PWK $P = 0.95$, Tukey's multiple comparisons test), in marked contrast to humans, where PRL seems to be released only around the time of ejaculation[15].

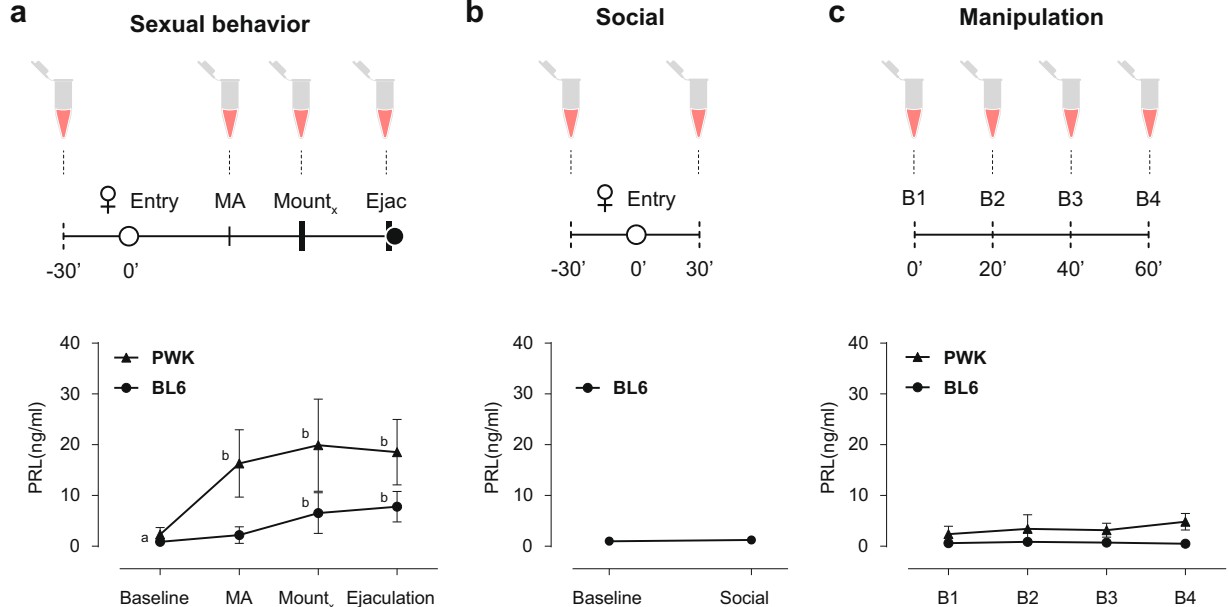

**Fig. 1 Prolactin is released during sexual behavior in male mice. a** Timeline for blood collection and [PRL]$_{blood}$ during sexual behavior (MA-mount attempt; BL6$_{Xmounts}$ = 5; PWK$_{Xmounts}$ = 3). RM one-way Anova for BL6 ($n = 8$) $F_{3,7} = 21.26$, $P < 0.0001$ and PWK ($n = 9$) $F_{3,8} = 17.18$, $P < 0.0001$, followed by Tukey's multiple comparison test $_{ab}P \leq 0.01$. **b** Timeline for blood collection and [PRL]$_{blood}$ during social behavior in BL6 males ($n = 15$) $P = 0.282$, two-tailed paired $t$-test. **c** Timeline for blood collection and [PRL]$_{blood}$ during repeated sampling in resting condition. RM one-way Anova for BL6 ($n = 8$) $F_{3,7} = 2.08$, $P = 0.18$ and PWK ($n = 8$) $F_{3,7} = 2.94$, $P = 0.11$. Data represented as mean ± SD.

Contrary to PWK males, which in the presence of a receptive female always engaged in sexual behavior and ejaculated, a large percentage of BL6 males never attempted to mount the female (15 out of 23, Supplementary Data 1). In such case, the session was aborted after 30 min of social interaction and a blood sample was collected (Fig. 1b). In this case, PRL levels of BL6 males did not differ from baseline (baseline $0.99 \pm 0.67$ vs social $1.25 \pm 0.63$; $P = 0.282$, paired $t$-test), further suggesting that PRL is only released in the context of a sexual interaction.

Because PRL is known to be released under stress[47] and to ensure that the changes observed in circulation are not a result from the blood collection procedure itself, all animals were initially habituated to the collection protocol in another cage, alone. To ensure that the habituation protocol was effective, in a separate experiment we measured PRL levels in the absence of any behavior. Four blood samples were collected 20 min apart from BL6 and PWK males in their home cage (Fig. 1c). In both cases, circulating PRL levels were not altered, ensuring that the observed increases were not caused by the manipulation (Bl6: $F_{3,7} = 2.08$, $P = 0.18$; PWK $F_{3,7} = 2.94$, $P = 0.11$, RM one-way ANOVA).

Together, these results demonstrate that PRL is released during sexual behavior in male mice, but not during a social interaction or due to the blood collection protocol, prompting us to examine the role of PRL release during sexual behavior.

**Acute prolactin release does not induce a refractory period-like state.** To investigate if the increase in circulating levels of PRL that occurs during mating is sufficient to decrease sexual activity, a hallmark of the PERP, we employed a pharmacological approach to acutely elevate PRL levels before the animals became sexually aroused and assess if the male mice behave as if in a PERP-like state. PRL is produced in specialized cells of the anterior pituitary, the lactotrophs, and its release is primarily controlled by dopamine originating from the hypothalamus[11]. Dopamine binds D2 receptors at the membrane of the lactotrophs, inhibiting PRL release. Suppression of dopamine discharge leads to disinhibition of lactotrophs, which quickly release PRL into circulation[48,49]. To acutely elevate PRL levels, we performed an intraperitoneal injection of the D2 dopamine receptor antagonist domperidone, which does not cross the blood–brain barrier[50,51], and measured PRL levels 15 min after the procedure. As expected, domperidone administration lead to a sharp rise in the levels of circulating PRL (BL6: baseline $0.4957 \pm 0.4789$ vs Domp $12.54 \pm 2.032$, $P < 0.0001$; PWK: baseline $4.82 \pm 1.645$ vs Domp $25.87 \pm 7.154$, $P < 0.0001$, paired $t$-test) (Fig. 2a) of similar magnitude to what is observed during copulation (ejaculation time point from Fig. 1a).

Therefore, on a separate experiment we investigated how domperidone-treated male mice behave with a receptive female. If PRL is sufficient to induce a PERP-like state, treated males should exhibit decreased sexual activity, which could be manifested in distinct manners, such as on the latency to initiate consummatory behavior or on the vigor of copulation. Each male from the two strains was tested twice, once with vehicle and another time with domperidone, in a counterbalanced manner and all the annotated behaviors are depicted over time on Fig. 2b, c (see "Methods" for details). Some BL6 males did not reach ejaculation in one of the sessions. A similar frequency was observed among unmanipulated animals (Supplementary Data 1). Despite differences in the dynamics of sexual behavior between BL6 and PWK males, administration of domperidone does not seem to affect sexual performance, as we could not detect any significant difference in the latency to start mounting the female, frequency of attempts to mount the female, time taken to ejaculate, or proportion of animals that reached ejaculation (Fig. 2d–g). Domperidone administration also does not seem to affect the dynamics of the sexual interaction across the session or within each mount (Fig. 2h, i, respectively) or other measures of sexual behavioral performance (Supplementary Data 2). Moreover, the majority of males reached ejaculation (latency to ejaculate since female entry: BL6:$16.2 \pm 7$ with a maximum of 62 min; PWK: $12.3 \pm 6.7$ with a

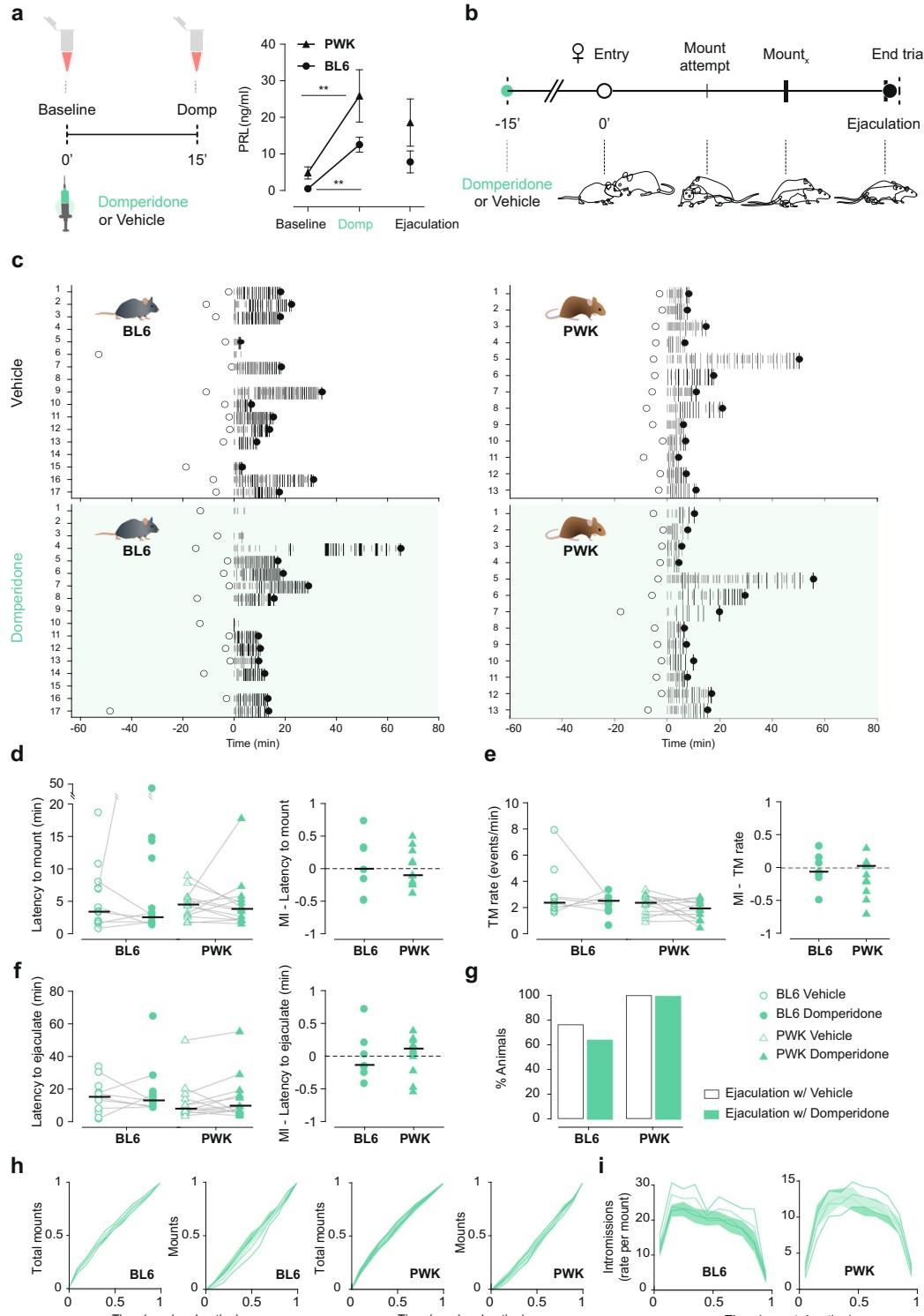

maximum of 59 min, Supplementary Data 2) within the time window where PRL levels remain elevated after domperidone administration[46].

In summary, domperidone administration, which causes an acute elevation of circulating PRL levels similar to what is observed at the end of copulation, does not have an inhibitory effect on any behavioral parameter related to sexual activity on the two strains of mice tested, that is, it does not induce a PERP-like state.

**Blocking prolactin release during copulation does not decrease the duration of the refractory period.** The release of PRL which is observed during sexual behavior has been proposed to be crucial in the establishment of the PERP[9]. To test this hypothesis, we acutely inhibited PRL release during sexual behavior by taking advantage of bromocriptine, a D2 receptor agonist. Bromocriptine's activation of D2 receptors on the lactotrophs' membrane blocks PRL release, a well-established procedure to inhibit the discharge of this hormone from the pituitary[34,52]. If PRL is

**Fig. 2 Acute prolactin release does not induce a refractory period-like state. a** Timeline for blood collection and [PRL]$_{blood}$ after Vehicle (Veh—black) or Domperidone (Domp—green) administration. Domp administration lead to a sharp rise in the levels of circulating PRL (Baseline vs Domp: BL6 $n = 8$, $P < 0.0001$; PWK: $n = 9 < 0.0001$, paired $t$-test) of similar magnitude of PRL release during copulation. Values from the ejaculation time point (Fig. 1a) were included as a reference. **b** Timeline for sexual behavior assay using sexually trained BL6 and PWK males pre-treated with Veh or Domp ($t = -15$ min). Each animal was tested twice, in a counterbalanced manner: once with Veh and once with Domp. **c** Raster plot aligned to the first consummatory event (first mount attempt or mount), representing the sexual behavior executed by the male, with mount attempts represented in small gray bars, mounts in long black bars (width correlated with mount duration), and ejaculation with a black circle. Time of female entry in the apparatus represented with an open circle. BL6 $n = 17$; PWK $n = 13$. Quantification of **d** latency to mount (first mount attempt or mount, BL6 $P = 1.0$, PWK $P = 0.6355$), **e** rate of total mounts (TM, mount attempts + mounts) (BL6 $P = 0.9375$, PWK $P = 0.3054$) and **f** latency to ejaculate (from **d**, BL6 $P = 0.8125$, PWK $P = 0.21631$). With BL6 represented by circles and PWK by triangles (Veh—open, Domp—solid). Each line represents data of an individual. Only animals that ejaculated in both sessions were considered in the statistics ($n_{BL6} = 7$, $n_{PWK} = 13$). Individuals that did not ejaculate in one of the trials are represented as unconnected dots (not used in statistics). MI [modulation index (Domp − Veh)/(Domp + Veh)] between the two conditions for both strains. Data presented as median ± M.A.D. (median absolute deviation with standard scale factor) following Wilcoxon rank-sum test. **g** Percentage of animals that reached ejaculation in the Veh (open) and Domp (solid) condition. **h** Cumulative distributions of total mounts and mounts along the behavioral assay. Histogram aligned to the first consummatory event with 0.1 min bins. Time normalized for the duration of the session from female entry to ejaculation. **i** Rate of intromissions executed during the mount. Histogram for all mounts of each session, aligned to beginning of the mount with 0.1 min bins. Time normalized for the duration of the mount.

indeed necessary for the establishment of the PERP, we expected that after ejaculation, drug-treated males would regain sexual activity faster than controls.

To test bromocriptine's efficiency in blocking PRL release during sexual behavior, we first injected a group of males with bromocriptine (or vehicle) and measured PRL levels at three time points: (i) before the drug or vehicle injection, (ii) before the female was inserted in the cage, and then (iii) after ejaculation (Fig. 3a). As shown in Fig. 3a, bromocriptine administration efficiently blocked PRL release in both subspecies of mice, since PRL levels after ejaculation are not different from baseline (BL6: B1 vs ejac, $P = 0.3$; B2 vs Ejac $P = 0.99$; PWK: B1 vs ejac, $P = 0.97$, B2 vs Ejac $P = 0.99$; Tukey's multiple comparisons test after RM two-way Anova).

In a separate experiment we then tested the effect of the pharmacological manipulation on the establishment of the PERP and its duration. In this case, the male and female were allowed to remain in the cage undisturbed for a period of up to 2 h after ejaculation. Each male from the two strains was tested twice, once with vehicle and a second time with bromocriptine, in a counterbalanced manner. Each session ended once the male performed the first attempt of copulation after ejaculation (red diamonds) or after 2 h if no attempt was made (Fig. 3b, see "Methods" for details). All the annotated behaviors are depicted over time in Fig. 3b, c (see "Methods" for details).

As shown in Fig. 3d, inhibiting PRL release during sexual behavior did not change the proportion of male mice of the two strains that reached ejaculation (Supplementary Data 1) or regained sexual activity in the 2 h after ejaculation (corresponding to the proportion of red diamonds in Fig. 3c). In addition, and contrary to what was expected, we observed a significant increase in the PERP of PWK males (Fig. 3e, veh: $21.7 \pm 4.18$ vs bromo: $35.4 \pm 16.3$, $P = 0.007$ by Wilcoxon signed rank test). Administration of bromocriptine also seems to affect the initial sexual arousal, as we could detect a decrease in the latency to start mounting the female (Fig. 3f, trend for B6 males and significant for PWK, veh: $4.06 \pm 4.35$ vs bromo: $1.93 \pm 1.13$, $P = 0.06$; and veh: $2.78 \pm 1.7$ vs bromo: $1.48 \pm 0.55$, $P = 0.01$, respectively, by Wilcoxon signed rank test). This observation was not due to an increase in activity/locomotion of the bromocriptine-treated males, as the average male speed before and after the female entry was not affected by the manipulation, nor the distance between the pair (Supplementary Data 3). Despite the locomotor activity being the same, the average male speed projected towards the female increased significantly for PWK males treated with bromocriptine as they moved in a goal-directed way, directionally towards the female (Supplementary Data 3).

Once consummatory behavior was initiated, control and bromocriptine-treated males exhibited similar levels of sexual performance, as we could not detect any difference in the frequency of attempts to mount the female or time taken to reach ejaculation (Fig. 3g, h). Other aspects of the sexual interaction were also not altered (Supplementary Data 4). Furthermore, bromocriptine administration does not seem to affect the dynamics of the sexual interaction across the session or within each mount (Fig. 3I, j, respectively).

In summary, blocking PRL release during copulation does not affect the proportion of animals that regain sexual activity within 2 h after ejaculation and contrary to what was expected, bromocriptine leads to an increase in the duration of the PERP of PWK males. Except for a decrease in the latency to start mounting, maintaining circulating PRL low, at levels similar to what is observed prior to the sexual interaction, does not affect any of the parameters of sexual performance analyzed in both strains of mice.

## Discussion

The PERP is highly conserved across species and is characterized by a general decrease in sexual activity after ejaculation[2]. The pituitary hormone PRL is released during copulation and has been put forward as the main player in the establishment of the PERP[9]. However, the involvement of PRL in the establishment and duration of the PERP is controversial and has not been formally tested[2]. Here we show that despite being released during copulation as previously shown in other taxa, PRL is neither sufficient nor necessary for the establishment of the PERP.

We first showed that PRL is released during copulation in male mice. The proportion of BL6 males that engaged in sexual behavior when tail blood is collected is lower than the proportion of animals that copulated in the rest of the experiments, suggesting that the procedure affects their sexual activity. However, once they start mating, all animals reached ejaculation. Importantly, once habituated, the procedure itself does not lead to PRL release. This opens up the possibility to perform such type of experiments using an "within-animal" design, a very important point particularly when there is a large interindividual variability, while decreasing the number of animals used.

Despite being released during sexual behavior in mice, PRL dynamics are quite different from what has been observed in humans. In men, PRL seems to only be released around the time of ejaculation[15,16], and only when ejaculation is achieved[16]. Indeed, the fact that PRL surge was only observed when ejaculation was achieved was one of the main results that lead to the idea that PRL may play a role in the acute regulation of sexual

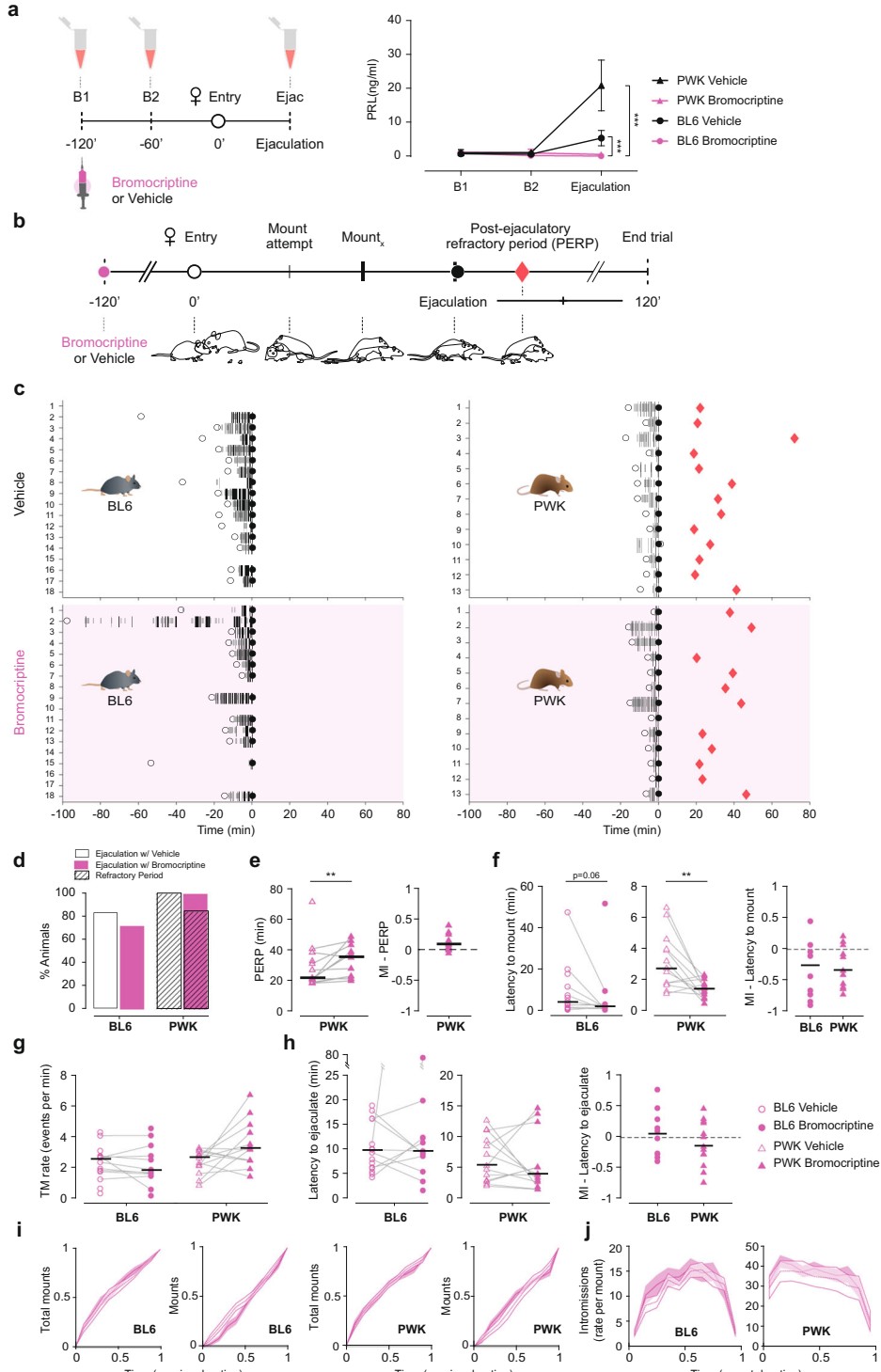

activity after orgasm in humans[53]. In contrast, in mice we observed an increase in circulating levels of PRL in sexually aroused PWK males and in BL6 males during the consummatory phase. The discrepancy between our results and the results published by others might be a result of the sampling procedure. Despite the fact that in human studies blood was continuously collected, PRL detection was performed at fixed time intervals and not upon the occurrence of particular events, such as ejaculation. Therefore, when averaging PRL levels across individuals, each participant might be in a slightly different internal state. Also, because PRL concentration is determined over fixed

intervals of time, it is difficult to pinpoint the PRL surge to the time of ejaculation (even though the human studies show that sexual arousal per se is not accompanied by an increase in PRL levels)[9]. To our knowledge, a single study assessed PRL levels during sexual behavior in male mice, stating that PRL is released after ejaculation[54]. In this case, blood was also continuously sampled at fixed intervals of time. In contrast, in our study the blood was collected upon the execution of particular events, such as the first MA, a pre-defined number of mounts and ejaculation. Thus, even though the intervals between PRL measurements are different for each mouse, we ensure that PRL levels are measured

**Fig. 3 Blocking prolactin release during copulation does not decrease the duration of the refractory period. a** Timeline for blood collection and [PRL]$_{blood}$ after Vehicle (Veh—black) or Bromocriptine (Bromo—pink) administration. RM two-way Anova with treatment (Veh or Bromo) as between subject's factor and time (B1, B2, and Ejac) as the within subject's factor, followed by Tukey's multiple comparison test: BL6 ($n = 5$ each) Treatment $F_{1,8} = 21.08$, $P = 0.0018$, Ejac$_{veh}$ vs Ejac$_{bromo}$ $P < 0.0001$; Time $F_{2,16} = 18.41$ $P < 0.0001$, B1$_{bromo}$ vs Ejac$_{bromo}$ $P = 0.3$; and PWK ($n_{veh} = 6$, $n_{bromo} = 4$) Treatment $F_{1,8} = 28.43$, $P = 0.0007$, Ejac$_{veh}$ vs Ejac$_{bromo}$ $P < 0.0001$; Time $F_{2,16} = 23.97$, $P < 0.0001$, B1$_{bromo}$ vs Ejac$_{bromo}$ $P = 0.97$. **b** Timeline for sexual behavior assay using sexually trained BL6 and PWK males pre-treated with Veh or Bromo ($t = -120$ min). Each animal was tested twice in a counterbalanced manner. **c** Raster plot aligned to ejaculation, representing the sexual behavior performed by the male, with mount attempts represented in small gray bars, mounts in long black bars (width correlated with mount duration), ejaculation with a black circle and PERP (latency to the first consummatory event (mount attempt or mount) after ejaculation represented with a red diamond). Time of female entry in the apparatus represented with an open circle. BL6 $n = 18$; PWK, $n = 13$. **d** Percentage of animals that reached ejaculation (solid color) and re-initiated the consummatory behavior after ejaculating (PERP, dashed color) in Veh (white) and Bromo (pink) conditions. Quantification of **e** PERP duration ($P = 0.007$), **f** latency to mount (first MA or mount) **(**BL6 $P = 0.0644$, PWK $P = 0.0132$), **g** rate of total mounts (TM, MA + mounts) (BL6 $P = 0.32$, PWK $P = 0.0681$) and **h** latency to ejaculate (from **f**, BL6 $P = 0.92$, PWK $P = 0.68$). With BL6 represented by circles and PWK by triangles (Veh—open, Bromo—solid). Each line represents data from an individual; only animals that ejaculated in both sessions were considered in the statistics ($n_{BL6} = 10$, $n_{PWK} = 13$). Individuals that did not ejaculate in one of the trials are represented as unconnected dots (not used in statistics). Data presented as median ± MAD (median absolute deviation with standard scale factor) following Wilcoxon rank-sum test. MI [modulation index (Bromo−Veh)/(Bromo + Veh)] between the two conditions for both strains. **P < 0.01. **i** Cumulative distributions of total mounts and mounts along the behavioral assay. Histogram aligned to the first consummatory event with 0.1 min bins. Time normalized for the duration of the session from female entry to ejaculation. **j** Rate of intromissions executed during the mount. Histogram for all mounts of each session, aligned to beginning of the mount with 0.1 min bins. Time normalized for the duration of the mount.

for all individuals in a similar internal state. Independently of the differences in the dynamics of circulating PRL levels, the increase we observe seems to be specific to a sexual encounter, since PRL levels in BL6 males that never attempt copulation remain unaltered from baseline.

In order to test if PRL by itself is sufficient to decrease sexual activity, we injected domperidone to induce an artificial PRL surge. In this case, the male mouse should behave like a male that just ejaculated: for example, exhibit longer latency to initiate the sexual interaction, which in the case of BL6 mice should take days. Even though domperidone administration leads to circulating levels of PRL that are similar to the ones observed at the end of a full sexual interaction, this manipulation did not cause any alteration in terms of sexual performance, as all behavioral parameters remained unaltered for both strains of mice. Although most likely other neuromodulatory systems were affected by our manipulation (via PRL), the fact that the domperidone manipulation did not cause a PERP-like state might still be due to the fact that the full repertoire of neuromodulators and hormones accompanying an ejaculation was not present[55]. Further experiments could test this idea by examining if combinations of different neuromodulators and hormones administered together with PRL can induce a PERP-like state.

Last, we asked if the elevation in PRL levels during sexual behavior is necessary for the establishment and duration of the PERP. For that we took a complementary pharmacological approach, where we injected bromocriptine, a D2 receptor agonist that temporarily inhibits the release of PRL. PRL levels after ejaculation in bromocriptine-treated males are similar to pre-copulatory levels. If PRL is indeed necessary to establish the PERP, we would expect a decrease in its duration that should easily be observed in the PWK males (since they regain sexual activity on average 30 min after ejaculation) or even in the BL6 (which take days). We observed a decrease in the latency to start mounting the female and, contrary to our expectation, a significant increase in the PERP duration of PWK males. We believe these effects may be mediated by the direct effect of bromocriptine, rather than an effect of PRL itself. First, baseline PRL levels are already very low in male mice and therefore the manipulation most likely did not affect them. Second, systemic administration of dopamine agonists has shown that anticipatory measures of sexual behavior are more sensitive to disruption than

are consummatory measures of copulation[56,57]. This agrees with our results, where we observed a significant decrease in the latency to initiate mounting with bromocriptine, while no other parameter of sexual performance was affected. Interestingly, bromocriptine-treated PWK males seem more ballistic in their approach to the female, suggesting a more goal-directed behavior towards the female. Bromocriptine (and domperidone) might also have an effect outside the central nervous system as D2 receptors are expressed in the human and rat seminal vesicles[58]. It is not known if direct manipulation of these receptors in the seminal vesicles has an impact on the PERP, which could explain our results.

In this study we investigated the role of PRL in the PERP of two different strains of mice that belong to the two main subspecies of house mouse, *Mus musculus musculus* (PWK) and *Mus musculus domesticus* (BL6), for three main reasons. As already presented, the two strains have very different PERP duration, widening the dynamic range of this behavioral parameter and increasing the probability of detecting an effect of the manipulations. Second, in addition to differences in PERP duration, BL6 and PWK males have very different sexual performance, reflected in several behavioral parameters as, for example, the number of mounts needed to reach ejaculation. Despite the differences (which were not explored as they are outside the scope of the present study), the effects of the pharmacological manipulations were similar across the two strains, causing either no effect or changing the behavior in the same direction (shorter latency to start mounting the female in bromocriptine-treated males). This is an important point that strengthens our conclusions. Finally, although fundamental for many present-day discoveries, the usage of the common inbred strains of mice comes at a cost, due to the limitations in their genetic background that sometimes leads to results that are specific to the strain of mouse used[59–61]. Wild-derived strains of mice are valuable tools that can complement the genetic deficiencies of classical laboratories strains of mice[45,62]. Despite the fact that larger numbers of animals are used (because experiments are repeated on each mouse strain), this approach is already routinely used in other fields, such as in immunological studies[63], providing greater confidence to the results obtained from the effect of pharmacological manipulations on behavior, for example.

As initially discussed, PRL can affect peripheral and central pathways that have been implicated in sexual performance and

motivation. Even though our manipulations indiscriminately affect both, our results refute the idea that PRL decreases sexual activity, the hallmark of the PERP. What could be the role of copulatory PRL? PRL release may be the "side-effect" of the neuromodulatory changes that occur during sexual behavior; this is merely the result of reduction in DA levels (DA inhibits PRL release) and/or the increase in oxytocin and serotonin (known stimulating factors of PRL release) instead of having the principal role in the establishment of PERP[64–66]. The fact that PRL levels are already elevated during sexual interaction in BL6 and PWK males further suggests that PRL cannot promote by itself reduced sexual activity, at least in male mice. Other studies point towards a role of PRL in the establishment of parental behavior[67–69]. New behavioral paradigms will be fundamental in unraveling this mystery.

## Methods

**Animals**. BL6 (*Mus musclus domesticus*, C57BL/6J) and Wild (*Mus musculus musculus*, PWD/PhJ and PWK/PhJ) mice were ordered from The Jackson Laboratories and maintained in our animal facility. Animals were weaned at 21 days and housed in same-sex groups in stand-alone cages (1284L, Techniplast, $365 \times 207 \times 140$ mm) with access to food and water ad libitum. Mice were maintained on a 12:12 light/dark cycle and experiments were performed during the dark phase of the cycle, under red dim light. All experiments were approved by the Animal Care and Users Committee of the Champalimaud Neuroscience Program and the Portuguese National Authority for Animal Health (Direcção Geral de Veterinária).

Females were kept house grouped and males were isolated before the sexual training. Both males and females were sexually experienced and habituated to be handled and to the assay routine. To enter the study, a male had to ejaculate three times (in four sessions). Males interacted with different females in each sexual encounter. Each manipulation was performed in a different group of males and, within each manipulation, experiments were conducted in parallel for both BL6 and PWK. Trials were conducted in the male home cage (1145T, Techniplast, $369 \times 156 \times 132$ mm) stripped from nesting, food, and water: covered with a transparent acrylic lid. The trial started with the entry of the female in the setup ($t = 0$ min).

**Ovariectomy and hormonal priming**. All females underwent bilateral ovariectomy under isoflurane anesthesia (1–2% at 1 L/min). After exposing the muscle with one small dorsal incision (1 cm), a small incision was made in the muscle wall, at the ovary level, on each side. The ovarian arteries were cauterized and both ovaries were removed. The skin was sutured, and the suture topped with iodine and wound powder. The animals received an i.p. injection of carpofen before being housed individually with food supplemented with analgesic (MediGel, 1 mg carprofen/2 oz cup) for 2 days recovery and then re-grouped in their home cages.

Female mice were primed subcutaneously 48 h before the assay with 0.1 ml estrogen (1 mg/ml, Sigma E815 in sesame oil) and 4 h before the assay with 0.1 ml progesterone (5 mg/ml, Sigma 088K0671 in sesame oil).

**Blood collection**. Tail-tip whole-blood samples were collected from the male tail, immediately diluted in PBS-T (PBS, 0.05% Tween-20), and frozen at –20 °C where it was stored until use[46].

To profile [PRL]$_{blood}$ during sexual behavior (Fig. 1a), baseline blood was collected 30 min before ($t = -30$ min) the female entry ($t = 0$ min). From this point on, blood collection was locked to the onset of specific behaviors: once the male did the first MA, after executing a fixed number of mounts (Mx) and after ejaculation (after the male exhibited the stereotypical shivering, falling to the side and decoupling from the female). We choose Mx = 5 for BL6 and Mx = 3 for PWK to ensure that the males would have sufficient sexual interaction without reaching ejaculation. Contrarily to PWK males that, in the presence of a receptive female, the majority engages in sexual behavior, BL6 do not. Thus, after 30 min interacting with the female without displaying sexual interest, we collected a blood sample and terminated the trail (Fig. 1b, social). Because blood collection is an invasive procedure and PRL is also released under stress, we evaluated if the manipulation itself could induce PRL release. For that we collected blood every 20 min for 1 h from males resting in their home cage (Fig. 1c).

Domperidone is a D2 dopamine receptor antagonist that was previously used to study the inhibitory tone of dopamine on PRL release from the pituitary, inducing a PRL peak 15 min after i.p. injection[46]. To test the magnitude of the PRL release of the two mouse strains under domperidone (Fig. 2a), we conducted a pilot study where we collected a blood sample before (baseline) and 15 min after Domp injection (20 mg/kg; Abcam Biochemicals). We opted to manipulate [PRL]$_{blood}$ through domperidone instead of injecting PRL directly to induce a PRL release similarly to a natural occurring instead of adding a recombinant form.

Bromocriptine is a D2 dopamine receptor agonist known to inhibit endogenous prolactin release[34]. To test its efficacy on blocking PRL release during sexual behavior (Fig. 3a), we conducted a second pilot study where the males were injected (100 µg bromo or vehicle) 2 h before the trial started. Blood samples were collected just before injection, 1 h after injection and after ejaculation.

**Prolactin quantification**. [PRL]$_{blood}$ quantification was done as previously described[46]. Briefly, a 96-well plate (Sigma-Aldrich cls 9018-100EA) was coated with 50 µl capture antibody anti-rat PRL (anti-rPRL-IC) (National Institute of Diabetes and Digestive and Kidney Diseases (NIDDK), AFP65191 (Guinea Pig), NIDDK-National Hormone and Pituitary Program (NHPP, TORRANCE, CA) at a final dilution of 1:1000 in PBS of the antibody stock solution, reconstituted in PBS as described in the datasheet (Na$_2$HPO$_4$ 7.6 mM; NaH$_2$PO$_4$ 2.7 mM; and NaCl 0.15 M; pH 7.4). The plate was protected with Parafilm® and incubated at 4 °C overnight in a humidified chamber. The coating antibody was decanted and 200 µl of blocking buffer (5% skimmed milk powder in PBS-T) was added to each well to block nonspecific binding. The plate was left for 2 h at room temperature on a microplate shaker. In parallel, a standard curve was prepared using a twofold serial dilution of Recombinant mouse Prolactin (mPRL; AFP-405C, NIDDK-NHPP) in PBS-T with BSA 0.2 mg/ml (bovine serum albumin; Millipore 82-045-1). After the blocking step, the plate was washed (three times for 3 min at room temperature with PBS-T), 50 µl of quality control (QC), standards or samples were loaded in duplicate into the wells, and incubated for 2 h at room temperature on the microplate shaker. The plate was washed, and the complex was incubated for another 90 min with 50 µl detection antibody (rabbit alpha mouse PRL, a gift from Patrice Mollard Lab) at a final dilution of 1:50,000 in blocking buffer solution. Following a final wash, this complex was incubated for 90 min with 50 µl horse-radish peroxidase-conjugated antibody (anti-rabbit, IgG, Fisher Scientific; NA934) diluted in 50% PBS and 50% blocking buffer. One tablet of *O*-phenylenediamine (Life Technologies SAS 00-2003) was diluted into 12 ml citrate-phosphate buffer pH 5, containing 0.03% hydrogen peroxide. One hundred microliters of this substrate solution was added to each well (protected from light), and the reaction was stopped after 30 min with 50 µl of 3 M HCl. The optical density from each well was determined at 490 nm using a microplate reader (SPECTROStar$^{Nano}$, BMG LABTECH). An absorbance at 650 nm was used for background correction.

A linear regression was used to fit the optical densities of the standard curve vs their concentration using samples ranging from 0.1172 to 1.875 ng/ml, using the former as the lower limit of detection. Appropriate sample dilutions were carried out in order to maintain detection in the linear part of the standard curve, and PRL concentrations were extrapolated from the OD of each sample. To control for reproducibility of the assay, trunk blood of males injected with domperidone was immediately diluted in PBS-T and pulled to be used as QC. Loading of the wells was done vertically left to right and QC was always loaded on the top row. The formula OD (Co, $t$) = OD (Ob) + $\alpha$ (QC).$t$ was used to correct the ODs for loading dwell time (OD: optical density, Co: corrected, $t$: well number, Ob: observed, $\alpha$: QC linear regression' $\alpha$). Coefficient of variability was kept to a maximum of 10%.

**Behavioral assays**. Each male underwent two trials: one with vehicle and one with drug (domperidone or bromocriptine). Administrations were counterbalanced between animals and spaced 7 days. In the first assay, for pharmacological induction of acute PRL release (Fig. 2b), the male was injected i.p. with domperidone or vehicle 15 min before the trial started ($t = -15$ min). Animals were allowed to interact until the male reached ejaculation or 1 h in the case the male did not display sexual behavior. Conversely, for pharmacological blockage of PRL release (Fig. 3b), a second group of males were pre-treated with bromocriptine or vehicle with a subcutaneous injection 2 h before the beginning of the trial ($t = -120$ min). Animals were allowed to interact a maximum of 2 h after the male reached ejaculation or 1 h in the case the male did not display sexual behavior. For the BL6 unmanipulated group, the female was added to the male home cage without disturbing the male (Supplementary Data 1). Animals were allowed to interact until the male reached ejaculation or 1 h in the case the male did not display sexual behavior.

**Behavior analysis**. The behavior was recorded from the top and side with pointgrey cameras (FL3- U3-13S2C-CS) connected to a computer running a custom Bonsai software[70]. Behavior was manually annotated using the open source program Python Video Annotator (https://pythonvideoannotator.readthedocs.io) and analyzed using Matlab. The number of MA (mount without intromission), mounts (mounts with intromission), latency to mount (first MA or mount), latency to ejaculation and PERP (latency between ejaculation and the next mount) was calculated. Total number of mounts (TM) was calculated as the sum of MA and mounts and TM rate was calculated as TM/latency to ejaculate. The percentage of animals that reached ejaculation and regain sexual activity under 2 h (refractory period) were also calculated. The modulation index (MI) was calculated as $(X_{drug} - X_{vehicle})/(X_{drug} + X_{vehicle})$. The centroid position and individual identity of each pair was followed offline using the open source program idtracker.ai[71] and used to calculate male velocity and interindividual distance with Matlab (Supplementary Data 3).

**Statistical analysis**. The statistical details of each experiment, including the statistical tests used and exact value of $n$, are detailed in each figure legend. Comparisons were always performed within the same strain and not across strains. Data related to prolactin quantification were analyzed using GraphPad Prism 7 software and presented as mean ± SD. For comparison within strain (Fig. 1a, c) an RM one-way Anova followed by a Tukey's multiple comparison test was used. Comparison of paired samples comparing two groups, statistical analysis was performed by using a paired-sample two-tailed $t$-test (Figs. 1b and 2a baseline-Domp). For the effect of treatment (veh or bromo) and time of sampling (B1, B2, and Ejac) (Fig. 3a) an RM two-way Anova followed by Tukey's multiple comparison test was used. Data related to animal behavior were analyzed with MATLAB R2019b and presented as median ± MAD (median absolute deviation with standard scale factor). Animals were randomized between treatments and comparison between the two conditions were done with Wilcoxon rank-sum test (Figs. 2d–f and 3e–h and Supplementary Data 2–4). Only animals that ejaculated in both sessions were included in the statistical analysis (Domperidone: $n_{BL6} = 7$, $n_{PWK} = 13$; bromocriptine: $n_{BL6} = 10$, $n_{PWK} = 13$). Significance was accepted at $P < 0.05$ for all tests.

**Reporting summary**. Further information on research design is available in the Nature Research Reporting Summary linked to this article.

## Data availability
All data generated to support the findings of this study are available from the corresponding author upon reasonable request.

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

## Acknowledgements

We thank Patrice Mollard and his team for welcoming S.V. in their facilities, teaching the uElisa technique, and for providing antibodies. We also express our gratitude to Francisco Romero for all the support with IdTracker, Gil Costa for the figure design, the Champalimaud Vivarium for their support with the wild derived animals, and the whole Lima lab for critical input. This work was supported by the Champalimaud Foundation, UIDB/04443/2020, LISBOA-01-0145-FEDER-022170, LISBOA-01-0145-FEDER-022231, ERC Consolidator Grant (772827, to S.Q.L.), Fundação para a Ciência e Tecnologia SFRH/BD/51011/2010 (to S.V.), and PhRMA Foundation Postdoctoral Fellowship in Informatics (to T.M.).

## Author contributions

S.Q.L. and S.V. designed the study. S.V. performed the experiments, annotation of the behavior, and IdTracker. T.M. wrote the Matlab code. S.V. analyzed the data with input from S.Q.L. and T.M. S.Q.L. and S.V. wrote the paper with contributions from others.

## Competing interests

The authors declare no competing interests.
