## [Peer Review File · Communications Biology]

Reviewers' comments:

Reviewer #1 (Remarks to the Author):

This paper examined whether the prolactin surge at mating governs the duration of the refractory period in male mice. This was tested by two methods: First, by increasing prolactin prior to mating and second, by blocking prolactin during mating. It has been a long-standing dogma of the field that prolactin regulates sexual satiety/refractory period in males. However, this assumption has not been backed up by strong evidence, even though it prevails the field as an established fact. This study challenges this dogma and shows that, in fact, prolactin does not affect the refractory period. The paper is further strengthened by it using a comparative model of mice that differ in sexual activity to show prolactin affects neither model. Overall, while the study is conceptually simple, it effectively provides evidence that prolactin does not have its presumed role in sexual satiety, thus provides important data that will have a big impact in this field.

I found the paper to be well-written and methods to be technically sound (and are the current best approaches for studying prolactin). There are a few unfortunate pitfalls of the study design which I found. While these don't completely compromise the project, they would have greatly strengthened the study's conclusions.

1. While the within-subjects design provides strong evidence that control and treated males do not behave differently I think this study would have really benefited from a second control group which was not bled or treated at all to show in fact the process of injections/tail bleeds did not affect sexual behavior. These treatments can be quite stressful, although this was not reflected in the prl measurements in Fig 1C. I was surprised that such a large percentage of B6 males (15/23 animals) did not show mounting behavior in the first experiment, perhaps suggesting that undergoing bleeding/injections may have interfered with the behavior somehow. This may be beyond the scope of a revision, but would increase the confidence of the current results. It could also be helpful if the authors had any additional info they could include re: if this a normal range of behavior when testing non-manipulated B6 males?

2. In the text, refractory period and sexual motivation (relating to after the first ejaculation) seem to be used interchangeably. However, I think the underlying motivational processes to engage in sexual behavior and physiological mechanisms that regulate the ability to perform sexual behaviors are two different things and so the authors should be careful about how they use this language. For instance, it may be more appropriate to state that they measured when males regain sexual activity, as opposed saying they measured when they regained sexual motivation, since the studies do not directly test which aspect of sexual behavior is being affected here.

3. It is unfortunate that the full suite of sexual behavior was not measured on the second mating attempt in the bromocriptine study. There are still a lot of unanswered questions about which aspects of sexual behavior/PERP prolactin could be potentially acting on - the latency to attempt mounting, latency to ejaculate, number of intromissions, etc. and while it was shown there was no difference in the time to the 2nd mount attempt, perhaps other aspects of the subsequent sexual encounter would have been affected. So, this it has somewhat limited the conclusions here. If the authors cannot retest this, then this should be addressed in the discussion.

4. Although they are the current gold standard methods, the use of bromocriptine and DOMP always comes with the questions about the specificity of prolactin manipulation. In the discussion, the authors states that other mechanisms may also contribute to sexual motivation such as serotonin, which is affected by both prolactin and bromocriptine. Although this may not be such a big deal since the authors did not find an effect here, it should be mentioned that these treatments do in fact affect these other systems, instead of claiming that these systems were not altered at all (would require other controls to show OT/5HT were not affected for example).

Minor comments

5. For the figures, a legend for the open/filled dots would be useful next to the figures.

6 For Fig 3C - should there be red diamonds on these graphs as well?

7. In the methods, could the authors state how much sexual experience males had before testing and for how long they were habituated for before the experiment?

8. I found it interesting that the behavior, while not different between treatment groups, was still quite variable within groups, even though they were treated the same way. This may actually provide more support that affecting prl did not change sexual behavior, but would be good to comment on this in the discussion somewhere.

Reviewer #2 (Remarks to the Author):

This manuscript describes a series of experiments to test the hypothesis that PRL mediates post-ejaculatory refractory period (PERP). The hypothesis that PRL plays a role in PERP has been circulating for decades, but much of the recent literature is consistent with PERP being a central, rather than endocrine, issue. As such, the premise of the research is not innovative. The authors' attempt to accept/reject this hypothesis is commendable, but a bit difficult due to multiple methodological issues. The manuscript also provides mostly negative results. In this case, the absence of proof may not proof of absence.

1) The major concern is that bromocriptine has off-target effects. The authors have also mentioned in the Discussion that bromocriptine can have central effects. In this case, how can the authors parse out the inhibition of PRL from the depressive central effects initiated by bromocriptine?

2) One concern is the invasive approach of obtaining blood samples during specific behaviors. The accepted techniques would be to continuously sample blood with an automatic system. These two methods should be compared side by side to ensure sample values are comparable.

3) The experimental designs for Fig. 2 and Fig. 3 are not parallel. It is unclear why Fig. 2 only measures sexual motivation and not PERP. Examining PERP would, in the case of PWK, reveal if Domp could delay the resumption of consummatory response and further strengthen authors' conclusion.

4) Bromocriptine reduced PRL secretion in Figure 3a, but the drug was administered at one set time point. The lag between the drug administration and behavioral onset could contribute to some variability. The time between the behavioral onset in relation to drug treatment should be discussed.

MINOR

1) Statistics were not performed properly. Data on repeated measures should be performed using 2-way RM ANOVA (instead of 1-way RM ANOVA; e.g., Figure 2A and 3A) following by post-hoc test.

2) The authors need to clarify behavioral signatures for ejaculation in mice, and how they captured mice in the act of ejaculation for blood sampling.

3) L38, change "to be" to "is"

- 4) L323, change "striped" to "stripped"
- 5) L356, change "trough" to "through"
- 6) The PRL ELISA limit of detection should be stated.

First and foremost, we would like to thank the reviewers for their constructive comments that have helped us improve the manuscript. Below please find a point-by-point reply to all comments raised.

Reviewer #1 (Remarks to the Author):

This paper examined whether the prolactin surge at mating governs the duration of the refractory period in male mice. This was tested by two methods: First, by increasing prolactin prior to mating and second, by blocking prolactin during mating. It has been a long-standing dogma of the field that prolactin regulates sexual satiety/refractory period in males. However, this assumption has not been backed up by strong evidence, even though it prevails in the field as an established fact. This study challenges this dogma and shows that, in fact, prolactin does not affect the refractory period. The paper is further strengthened by it using a comparative model of mice that differ in sexual activity to show prolactin affects neither model. Overall, while the study is conceptually simple, it effectively provides evidence that prolactin does not have its presumed role in sexual satiety, thus provides important data that will have a big impact in this field.

We thank the reviewer for their overall appreciation of the paper.

I found the paper to be well-written and methods to be technically sound (and are the current best approaches for studying prolactin). There are a few unfortunate pitfalls of the study design which I found. While these don't completely compromise the project, they would have greatly strengthened the study's conclusions.

1. While the within-subjects design provides strong evidence that control and treated males do not behave differently I think this study would have really benefited from a second control group which was not bled or treated at all to show in fact the process of injections/tail bleeds did not affect sexual behavior. These treatments can be quite stressful, although this was not reflected in the prl measurements in Fig 1C. I was surprised that such a large percentage of B6 males (15/23 animals) did not show mounting behavior in the first experiment, perhaps suggesting that undergoing bleeding/injections may have interfered with the behavior somehow. This may be beyond the scope of a revision, but would increase the confidence of the current results. It could also be helpful if the authors had any additional info they could include re: if this a normal range of behavior when testing non-manipulated B6 males?

We thank the reviewer for raising this issue. Indeed, the percentage of males that copulated in the first experiment (where tail blood was collected across the behavioral session) is lower when compared to the percentage of males that copulated in the other experiments (injection of Domp, Bromo or vehicle). This is most likely caused by the procedure itself, which is unavoidable in order to collect tail blood. We have included a comment in the discussion regarding this issue (Line #236). As the reviewer pointed out, despite the fact that the transition from appetitive to consummatory behavior was affected by the procedure, PRL levels were not altered by the blood collection after habituation. Therefore, we feel confident that the PRL elevation is related to the execution of the sexual behavior. Despite its negative impact on behavior, it gave us access to

the “perfect” controls, since non-copulating animals had a full social interaction with females, but we observed no increase in serum PRL.

Line # 236 The proportion of BL6 males that engaged in sexual behavior when tail blood is collected is lower than the proportion of animals that copulated in the rest of the experiments, suggesting that the procedure affects their sexual activity.

Regarding the Domp/Bromo/Vehicle injections, the proportion of animals that did not mate on a single session is around 30%. This is within the normal values that we observed in the lab after many years of sexual behavior experiments with BL6. As the reviewer suggested, we added the results of a new control experiment, with unmanipulated BL6 males. These animals were sexually trained like all other males included in the manuscript and the proportion of ejaculating individuals is similar to the injected ones. For that we added a new supplemental table (Supp Table 1) that includes the proportion of ejaculating males across all experiments, including the new control data set.

2. In the text, refractory period and sexual motivation (relating to after the first ejaculation) seem to be used interchangeably. However, I think the underlying motivational processes to engage in sexual behavior and physiological mechanisms that regulate the ability to perform sexual behaviors are two different things and so the authors should be careful about how they use this language. For instance, it may be more appropriate to state that they measured when males regain sexual activity, as opposed to saying they measured when they regained sexual motivation, since the studies do not directly test which aspect of sexual behavior is being affected here.

We thank the reviewer for the comment, we completely agree with it and changed the manuscript accordingly.

3. It is unfortunate that the full suite of sexual behavior was not measured on the second mating attempt in the bromocriptine study. There are still a lot of unanswered questions about which aspects of sexual behavior/PERP prolactin could be potentially acting on - the latency to attempt mounting, latency to ejaculate, number of intromissions, etc. and while it was shown there was no difference in the time to the 2nd mount attempt, perhaps other aspects of the subsequent sexual encounter would have been affected. So, this has somewhat limited the conclusions here. If the authors cannot retest this, then this should be addressed in the discussion.

We agree with the reviewer that it would have been interesting to continue monitoring the sexual performance of the Bromo treated males once they regain sexual activity after the first ejaculation. However, because our main goal was to investigate if PRL is needed to establish the PERP (and its duration), unfortunately we stopped the experiment after the first mount attempt and we do not have that data.

4. Although they are the current gold standard methods, the use of bromocriptine and DOMP always comes with the questions about the specificity of prolactin manipulation. In the discussion, the authors state that other mechanisms may also contribute to sexual motivation such as serotonin, which is affected by both prolactin and bromocriptine. Although this may not be such a big deal since the authors did not find an effect here, it should be mentioned that these treatments do in fact affect these other systems, instead of claiming that these systems were not altered at all (would require other controls to show OT/5HT were not affected for example).

We completely agree with the reviewer that what we wrote in the discussion could be misleading and we changed the text accordingly (Line # 272).

Line # 272 Although most likely other neuromodulatory systems were affected by our manipulation (via PRL), the fact that the domperidone manipulation did not cause a PERP-like state might still be due to the fact that the full repertoire of neuromodulators and hormones accompanying an ejaculation was not present.

Minor comments

5. For the figures, a legend for the open/filled dots would be useful next to the figures.

For sake of clarity, we added a legend next to the raster plots, indicating what each graphical element represents.

Line # 687 Figure 2

Line # 714 Figure 3

6. For Fig 3C - should there be red diamonds on these graphs as well?

The red diamonds represent the first mount after ejaculation. We added a note in the main text to make it clearer (Line # 196 and Line # 201). They only exist in the PWK males, as none of the BL6 males attempted copulation within the time allowed (2 hours after ejaculation, Fig3D).

Line # 196 Each session ended once the male performed the first attempt of copulation after ejaculation (red diamonds) or after two hours if no attempt was made (Fig. 3b, see Methods for details).

Line # 201 As shown in Fig. 3d, inhibiting PRL release during sexual behavior did not change the proportion of male mice of the two strains that reached ejaculation (also see Supplementary Table1) or regained sexual activity in the two hours after ejaculation (corresponding to the proportion of red diamonds in Fig3b).

7. In the methods, could the authors state how much sexual experience males had before testing and for how long they were habituated for before the experiment?

Following the suggestion of the reviewer, we added that information to the methods section (Line # 343).

Line # 343 Both males and females were sexually experienced and habituated to be handled and to the assay routine. To enter the study, a male had to ejaculate 3 times (in 4 sessions). Males interacted with different females in each sexual encounter. Each manipulation was performed in a different group of males and, within each manipulation, experiments were conducted in parallel for both BL6 and PWK.

8. I found it interesting that the behavior, while not different between treatment groups, was still quite variable within groups, even though they were treated the same way. This may actually provide more support that affecting prl did not change sexual behavior, but would be good to comment on this in the discussion somewhere.

We thank the reviewer for this comment. Indeed, there is a lot of individual variability in sexual performance despite that all males are treated in the same manner. We did not approach the difference in sexual performance across individuals and between the two strains of mice (besides the difference in PERP), because we felt it would make the storyline more complex and distract the reader from the main points. However, even though their sexual behavior is very different, the results of both manipulations are very similar in both strains, strengthening the point that PRL is not involved in the establishment of the PERP. We added a comment regarding the strain difference in the discussion (Line # 307).

Line # 307 Second, in addition to differences in PERP duration, BL6 and PWK males have very different sexual performance, reflected in several behavioral parameters as for example the number of mounts needed to reach ejaculation. Despite the differences (which were not explored as they are outside the scope of the present study), the effects of the pharmacological manipulations were similar across the two strains, causing either no effect or changing the behavior in the same direction (shorter latency to start mounting the female in bromocriptine treated males). This is an important point that strengthens our conclusions.

Reviewer #2 (Remarks to the Author):

This manuscript describes a series of experiments to test the hypothesis that PRL mediates post-ejaculatory refractory period (PERP). The hypothesis that PRL plays a role in PERP has been circulating for decades, but much of the recent literature is consistent with PERP being a central, rather than endocrine, issue. As such, the premise of the research is not innovative.

We thank the reviewer for the comment; indeed, the hypothesis has been circulating for years without formal testing. Also, as the reviewer points out, it is true that for a long time the PERP was mostly thought to be a “peripheral” problem and only more recently the involvement of the central nervous system has been brought into attention. Nevertheless, even if “recent literature is consistent with PERP being a central, rather than endocrine, issue”, to the best of our knowledge, a direct test of the role of PRL in PERP had not been performed before rendering our study as a novel finding worth being reported. Furthermore, PRL receptors are widely expressed in the brain, in many regions that are involved in the processing of conspecific cues and control of sexual behavior. Therefore, besides its purely endocrine effect, PRL can also alter the activity of the central nervous system related to arousal, motivation and sexual performance. We think we did not make this point clear enough in the introduction and for that we made some changes in the text (Line # 51); stating also in the discussion that our manipulations lead to changes in PRL levels that affect all systems (from endocrine to central effects) that respond to this hormone (Line #295).

Because there was no effect in sexual behavior as expected, our interpretation is that with our results we can refute that prolactin, either through peripheral or central effects, establishes the refractory period.

Line # 51 PRL is primarily produced and released into the bloodstream from the anterior pituitary^{11,28} and consistent with its functional diversity, PRL receptors are found in most tissues and cell types of the body^{29,30}. Therefore, PRL may depress sexual activity directly, via PRL receptors present in the male reproductive tract. In fact, PRL has been shown to impact the function of accessory sex glands and to contribute to penile detumescence³¹. PRL can also affect central processing, as it can reach the central nervous system either via circumventricular regions lacking a blood-brain barrier³² or via receptor-mediated mechanisms³³, binding its receptors which have widespread distribution, including in the social brain network³⁴. Hence, circulating PRL can impact the activity of neuronal circuits involved in the processing of socio-sexual relevant and thus sexual performance. Circulating PRL reaches the central nervous system on a timescale that supports the rapid behavioral alterations that are observed immediately after ejaculation (in less than 2 minutes)³⁵. Through mechanisms that are not yet well established, PRL is known to elicit fast neuronal responses³⁶ besides its classical genomic effects³⁷. In summary, circulating PRL can impact several systems involved in sexual behavior on a time scale compatible with the establishment of the PERP.

Line # 295 Bromocriptine (and domperidone) might also have an effect outside the central nervous system as D2 receptors are expressed in the human and rat seminal vesicles⁶⁴. It is not known if direct manipulation of these receptors in the seminal vesicles has an impact on the PERP, which could explain our results.

The manuscript also provides mostly negative results. In this case, the absence of proof may not proof of absence.

We agree with the reviewer that we mostly present negative results. However, even though our manipulations lead to significant changes in prolactin levels in circulation, as expected (Domp leads to prolactin increase and Bromo to clearing of prolactin from circulation), these did not cause significant alterations in sexual behavior as it was hypothesized. We also acknowledge that “not being different” is not the same as stating “they are the same”.

1) The major concern is that bromocriptine has off-target effects. The authors have also mentioned in the Discussion that bromocriptine can have central effects. In this case, how can the authors parse out the inhibition of PRL from the depressive central effects initiated by bromocriptine?

Indeed, as the reviewer points out, and as discussed, bromocriptine might have some potential off target effects that we cannot avoid. We think this would be a problem if in fact we observed the expected result: a decrease in the PERP duration, prompting us to perform additional manipulations to test specificity of our manipulation. However, this was not the case.

Instead, we observed some other effects of the bromocriptine treatment, which may or may not be PRL-mediated and which we think are not relevant for our hypothesis:

a) Males seem to be more sexually aroused, as they start mounting the female faster; Because PRL levels in bromocriptine-treated animals are low, similar to what is found in unmanipulated males, this effects does not seem to be PRL-dependent; we think it is most likely due to the central action of bromocriptine in brain D2 receptors which are known to promote sexual arousal (even though then all other metrics of sexual behavior are unaltered);

b) Bromocriptine treated PWK males have a longer PERP. If we take into consideration the first effect (shorter latency to start mounting) we would expect that if bromocriptine has a central effect, it should be in the same direction, this is, shortening the latency to start mounting after ejaculation. What we observe is the opposite, a lengthening of the PERP. Therefore, a putative direct effect of bromocriptine should not mask an effect caused by the absence of PRL. We added a comment in the discussion regarding this last point (Line #298).

Line # 298 In fact, if we take into consideration the effect bromocriptine has on the beginning of the session (shorter latency to start mounting) we would expect that any central effect of this drug after ejaculation should be in the same direction, this is, shortening the PERP. What we observe is the opposite, a lengthening of the PERP. Therefore, it is unlikely that bromocriptine administration masked an effect caused by the absence of PRL.

2) One concern is the invasive approach of obtaining blood samples during specific behaviors. The accepted techniques would be to continuously sample blood with an automatic system. These two methods should be compared side by side to ensure sample values are comparable.

We agree with the reviewer that a less invasive blood monitoring technique could be beneficial to determine PRL levels. However, despite being less invasive, those methods would still likely affect behavioral performance, because the animal has to be connected via a tube, which would constrain its movements. Also, these less invasive techniques can probably be used in BL6 animals, but it would probably be unfeasible to use them on the PWK males, which are much smaller. Finally, despite the fact that we observed an impact on sexual behavior performance on the blood analysis experiments, we feel confident that the PRL levels recorded are specific to sexual behavior, since we did not observe an increase in animals that only performed social interactions or when we performed blood sampling alone. Therefore, even when considering the experimental limitations of the method used, we are extremely confident with our interpretation of the results, and we do not think that we would gain information by using a continuous blood sample technique.

3) The experimental designs for Fig. 2 and Fig. 3 are not parallel. It is unclear why Fig. 2 only measures sexual motivation and not PERP. Examining PERP would, in the case of PWK, reveal if Domp could delay the resumption of consummatory response and further strengthen authors' conclusion.

As the reviewer pointed out, experiments 2 and 3 are not parallel, both in design and the way they are presented in the manuscript. This derives from the fact that each experiment tests a different hypothesis. We agree that it could be interesting to study the PERP of animals treated with Domp, but the main question was to check if acute high levels of PRL, before a sexual interaction, could drive the animal into a PERP-like state. That was not the case for both strains of mice, as the Domp treatment had no effect in any metrics of sexual performance. Also, because the Domp treatment leads to a surge of PRL which is within the range of what is observed after ejaculation, we should not expect to observe any difference in PERP duration.

4) Bromocriptine reduced PRL secretion in Figure 3a, but the drug was administered at one set time point. The lag between the drug administration and behavioral onset could contribute to some variability. The time between the behavioral onset in relation to drug treatment should be discussed.

We thank the reviewer for the comment, and it is true that animals start mounting and ejaculate at different times after bromocriptine administration. However, independently of when they started the behavior or ejaculated, we always obtained the same result: very low levels of PRL in circulation after ejaculation in bromocriptine treated animals. So, despite the behavioral jitter, there is no variability in PRL levels and therefore we think that cannot explain the results obtained. Also, the latency to ejaculate in bromocriptine treated animals as shown in Fig. 3h is not different from vehicle injected animals, supporting the observation that different levels of PRL are not setting the time for ejaculation.

MINOR

1) Statistics were not performed properly. Data on repeated measures should be performed using 2-way RM ANOVA (instead of 1-way RM ANOVA; e.g., Figure 2A and 3A) following by post-hoc test.

In this study we took advantage of two different subspecies of mice that have different PERP duration to evaluate the effect of PRL released during mating in the establishment of the refractory period. The scope of the study was to test the effect of the manipulations in the two models. For this matter, all analysis was performed between conditions for the same subspecies (drug-vehicle) and not across subspecies (BL6-PWK).

In Fig. 2a baseline and Domp conditions were compared using a two-tailed paired t test for each strain as the values were obtained from the same animals. As expected, domperidone administration leads to a sharp rise in the levels of circulating PRL. For comparison, we included

the values from the ejaculation time point from Fig. 1a as a reference to show that domperidone induced PRL surge is of similar magnitude to what is observed during copulation. We have clarified our analyses in the figure legend and main text (Line #150 and Line # 689).

Line # 687 Figure 2

Line # 150 As expected, domperidone administration lead to a sharp rise in the levels of circulating PRL (BL6: baseline 0.4957 ± 0.4789 vs domp 12.54 ± 2.032 , $P < 0.0001$; PWK: baseline 4.82 ± 1.645 vs domp 25.87 ± 7.154 ; < 0.0001 , Paired t test) (Fig. 2a) of similar magnitude to what is observed during copulation (ejaculation time point from Fig. 1a).

Line # 689 **a** Timeline for blood collection and $[PRL]_{\text{blood}}$ after Vehicle (black) or Domperidone (Domp, green) administration. Domp administration lead to a sharp rise in the levels of circulating PRL, (baseline vs domp: BL6 $n = 8$, $P < 0.0001$; PWK: $n = 9 < 0.0001$, Paired t test) of similar magnitude of PRL release during copulation. Values from the ejaculation time point (Fig. 1a) were included as a reference

In Fig 3a, although by lapse we described in the material and methods section as been performed a paired-sample two-tailed t test, in the figure legend (Line # 724) and main text (Line # 189) correctly describes the use of a RM two-way Anova with treatment (veh or bromo) as between subject's factor and time (B1, B2 and Ejac) as the within subject's factor, followed by Tukey's multiple comparison test.

2) The authors need to clarify behavioral signatures for ejaculation in mice, and how they captured mice in the act of ejaculation for blood sampling.

We thank the reviewer for the comment. We edited the text accordingly in the main text (Line # 108) and in the methods (Line # 369). The blood was collected immediately after the male detached from the female.

Line # 108 and after ejaculation (*ejaculation*, after the male exhibited the stereotypical shivering, falling to the side and separating from the female) (Fig.1a, please see Methods for details).

Line # 369 and after ejaculation (after the male exhibited the stereotypical shivering, falling to the side and decoupling from the female).

3) L38, change “to be” to “is”

4) L323, change “striped” to “stripped”

5) L356, change “trough” to “through”

We thank the reviewer for all the typos found, we made the appropriate changes.

6) *The PRL ELISA limit of detection should be stated.*

We thank the reviewer for the comment. Therefore, the limit of detection was included in the material and methods section (Line # 416).

Line # 416 A linear regression was used to fit the optical densities of the standard curve vs their concentration using samples ranging from 0.1172ng/ml to 1.875ng/ml, using the former as the lower limit of detection

REVIEWERS' COMMENTS:

Reviewer #1 (Remarks to the Author):

The revised manuscript has addressed my concerns/comments and I recommend that it be accepted.

I only found a couple of small things which should be changed before acceptance:

-Do you have any other information re: how long DOMP increases PrI for (beyond 15 mins)? (i.e. was it still elevated during sexual behavior)?

-line 172 – should this read “that is” instead of “this is”?

Fig 2g – figure legends states “Ejaculation with bromocriptine”... should be DOMP?

Reviewer #2 (Remarks to the Author):

The authors have adequately addressed most of the concerns. It is, however, still problematic that bromocriptine can have off-target central effects. The authors have addressed this obliquely but not experimentally. There is one issue that needs to be addressed.

The authors indicated that “Bormocriptine treated PWK males have a longer PERP. If we take into consideration the first effect (shorter latency to start mounting) we would expect that if bromocriptine has a central effect, it should be the same direction...”. It is unclear what is the basis for this explanation. PERP likely involves different mechanisms and neuronal populations.

The authors are conflating PERP and mounting, and these two phenomena need to be separated in the revision.

First and foremost, we would like to thank the reviewers once again for their constructive comments. Below please find a point-by-point reply to the comments raised by the reviewers.

Reviewer #1 (Remarks to the Author):

The revised manuscript has addressed my concerns/comments and I recommend that it be accepted.

We thank the reviewer for the positive assessment of the manuscript.

I only found a couple of small things which should be changed before acceptance:

1. Do you have any other information re: how long DOMP increases Prl for (beyond 15 mins)? (i.e. was it still elevated during sexual behavior)?

We thank the reviewer for bringing up this issue. In fact, we know that PRL levels start to decrease 60 minutes after domperidone administration. Almost all male mice ejaculated in less than 60 minutes (after female entry) and therefore we feel confident that PRL levels were high (and constant) during the whole sexual interaction. We added this information to the revised manuscript.

Line 166:

Moreover, the majority of males reached ejaculation (latency to ejaculate since female entry: BL6:16.2 ± 7 with a maximum of 62 min; PWK: 12.3 ± 6.7 with a maximum of 59 min, Supplementary Fig. 1e) within the time window where PRL levels remain elevated after domperidone administration⁴⁶.

2. line 172 – should this read “that is” instead of “this is”?

3. Fig 2g – figure legends states “Ejaculation with bromocriptine”... should be DOMP?

We thank the reviewer for identifying these two typos, which are now corrected in the revised version.

Reviewer #2 (Remarks to the Author):

The authors have adequately addressed most of the concerns. It is, however, still problematic that bromocriptine can have off-target central effects. The authors have addressed this obliquely but not experimentally. There is one issue that needs to be

addressed.

- The authors indicated that “Bromocriptine treated PWK males have a longer PERP. If we take into consideration the first effect (shorter latency to start mounting) we would expect that if bromocriptine has a central effect, it should be the same direction...”. It is unclear what is the basis for this explanation. PERP likely involves different mechanisms and neuronal populations.

The authors are conflating PERP and mounting, and these two phenomena need to be separated in the revision.

We agree with the reviewer that the PERP and mounting per se most likely involve different mechanisms and that our sentence regarding this issue was confusing. We were trying to explain that bromocriptine’s effect in the beginning of the trial (shorter latency to mount the female which is the same behavioral event that we use to determine the PERP duration after ejaculation) did not occur after ejaculation, since we did not observe a decrease in the latency to start mounting the female. We prefer to remove the sentence from the discussion, as it does not add any relevant information.